# *Pseudomonas aeruginosa* Bacteriophages and Their Clinical Applications

**DOI:** 10.3390/v16071051

**Published:** 2024-06-29

**Authors:** Elaheh Alipour-Khezri, Mikael Skurnik, Gholamreza Zarrini

**Affiliations:** 1Department of Animal Biology, Faculty of Natural Sciences, University of Tabriz, Tabriz 51368, Iran; elahealp1999@gmail.com; 2Human Microbiome Research Program, and Department of Bacteriology and Immunology, Faculty of Medicine, University of Helsinki, 00290 Helsinki, Finland; 3Microbial Biotechnology Research Group, University of Tabriz, Tabriz 51368, Iran

**Keywords:** *Pseudomonas aeruginosa*, bacteriophage, phage therapy, cystic fibrosis

## Abstract

Antimicrobial resistance poses a serious risk to contemporary healthcare since it reduces the number of bacterial illnesses that may be treated with antibiotics, particularly for patients with long-term conditions like cystic fibrosis (CF). People with a genetic predisposition to CF often have recurrent bacterial infections in their lungs due to a buildup of sticky mucus, necessitating long-term antibiotic treatment. *Pseudomonas aeruginosa* infections are a major cause of CF lung illness, and *P. aeruginosa* airway isolates are frequently resistant to many antibiotics. Bacteriophages (also known as phages), viruses that infect bacteria, are a viable substitute for antimicrobials to treat *P. aeruginosa* infections in individuals with CF. Here, we reviewed the utilization of *P. aeruginosa* bacteriophages both in vivo and in vitro, as well as in the treatment of illnesses and diseases, and the outcomes of the latter.

## 1. Introduction

Antibiotics are currently the major treatment for bacterial infections [1]. Numerous issues—among others, the development of drug-resistant bacteria, immune system suppression, drug remnants in animal products, and environmental contamination—have arisen due to the overuse of antibiotics. Bacteriophages (phages for short) are viruses that have no harmful effects on human or animal cells but can infect and kill bacteria. To treat bacterial infections, they can be used alone or together with antibiotics [2].

The United Nations (UN) predicted that in 2050, there would be roughly 10 million fatalities annually due to infections caused by multidrug resistant (MDR) bacteria. Therefore, it is crucial to discover fresh treatment approaches as the efficiency of conventional antibiotics fade away. The use of phages can be one of these remedies. Phages comprise the largest and most varied biological group on Earth. Their host specificity and persistence in natural systems are very high [3]. Because they must have a bacterial host to propagate, phages are widespread and can be found almost anywhere in the biosphere where bacteria are present [4]. Additionally, phages are effective therapeutic agents [5]. Phage treatment, in contrast to antibiotics, specifically eradicates only the host bacteria without affecting the non-host bacteria [6]. The number of phages is correlated with the number of their host bacteria; as a consequence, when the numbers of the targeted pathogens are reduced, the number of phages is likewise reduced, preserving microbial stability and variety [5].

Interest in bacteriophages has increased recently because using them to inhibit bacteria may be a secure, efficient, and all-natural alternative to antibiotic therapy. Bacteriophages were used in medicine prior to the discovery of antibiotics [7]. Scientific and clinical research into the use of bacteriophages was neglected because antibiotics were initially very effective at treating infections. Only the sharp rise in the prevalence of antibiotic-resistant strains sparked renewed interest in phage therapy. *Pseudomonas aeruginosa* is a metabolically flexible Gram-negative opportunistic bacterium that can infect a wide range of hosts and is found in a wide range of biotic and abiotic settings, such as water, soil, plants, and animals. It is a major cause of death, especially for people with immunosuppression, burns, severe wounds, and cystic fibrosis (CF) [8,9,10]. Patient morbidity and mortality rise when resistant forms of nosocomial infections arise. Thus, there is an urgent need for a novel and successful treatment for infections brought on by MDR *P. aeruginosa* strains. In this article, we review recent investigations on *P. aeruginosa* -specific phages, carried out both in vitro and in vivo, and their efficiency in the treatment of *Pseudomonas* infections.

## 2. Structure and Taxonomy of *P. aeruginosa* Phages

Phages differ in size, complexity, genetic make-up, and morphology [11]. The historical taxonomy of bacteriophages, which dates back to David Bradley in the 1960s and 1970s, comprising morphologically diverse varieties, such as filamentous, icosahedral, and tailed phages that were further separated depending on the nature of their nucleic acid (single- or double-stranded RNA or DNA) [12]. The morphology-based families *Myoviridae* (with a contractile tail), *Podoviridae* (with a short, stubby tail), and *Siphoviridae* (with a long, flexible tail) as well as the order *Caudovirales* were abolished in 2023 by the Bacterial Viruses Subcommittee of the International Committee on Taxonomy of Viruses (ICTV) [13]. The old classification was replaced by the class *Caudoviricetes*, under which new orders, families, subfamilies and genuses were established. This adjustment was made, as the original morphologic categorization did not adequately capture the common evolutionary histories of phages [14,15]. Caudoviricetes is a class of viruses known as tailed bacteriophages (cauda is Latin for “tail”). The virus particles have a distinct shape; each virion has an icosahedral head that contains the viral genome and is attached to a flexible tail by a connector protein [16].

*Pseudomonas*-specific phages were initially discovered in the middle of the 20th century, and various in vitro investigations have recently been carried out to assess their efficacy against infections caused by *P. aeruginosa*, including MDR strains [17]. Altogether, 137 phages that targeted *P. aeruginosa* were sequenced and published in March 2019 [17], and the number has increased to over 3000 since then. Most *Pseudomonas* phages are tailed, possess double-stranded DNA (dsDNA) genomes, have either myovirus, podovirus or siphovirus morphology, and are, thus, members of the class *Caudoviricetes*. Therapeutically acceptable phages should be strictly lytic and devoid of harmful genes that encode, for example, virulence factors, toxins, or antibiotic resistance.

The characteristics of phages that have been used for therapeutic experiments are shown in Table 1. They are very diverse. The type phage phiKMV and its relatives in the genus Phikmvvirus are known to be highly virulent phages that produce large clear plaques (diameter 3–15 mm) on susceptible hosts. They have a head of about 60–65 nm in diameter and a non-contractile tail with 6 prominent tail spikes [18]. Nipunavirus phages have an icosahedral head and a flexible tail, with genome sizes of about 58 kilobasepairs (kb) [19]. Casadabanvirus phages have a small icosahedral head of about 40 nm in diameter. The non-contractile tail is about 190 nm in length. Some members of the genus are temperate [20]. The capsid of Litunavirus phages measures approximately 60–85 nm in diameter, and it has a short 10–40 nm tail attached at the portal vertex [21]. Viruses in Bruynoghevirus are non-enveloped, with an icosahedral head (63 nm) and a short tail (12 × 8 nm) that has six prominent tail spikes [22]. Phikzviruses are non-enveloped jumbo-phages whose heads have a relatively large diameter of about 140 nm to allow for packaging of the >200 kb genome. The tails of these phages are around 160 nm long and 35 nm in diameter [23]. The cystovirus phiYY has an icosahedral particle covered by an outer layer of protein and a lipid envelope, with a diameter of about 85 nm, and its dsRNA genome consists of three fragments and encodes twelve proteins [24]. 

## 3. Infectious Diseases with *P. aeruginosa* as the Principal Causative Agent 

The National Healthcare Safety Network (NHSN) at the Centers for Disease Control and Prevention (CDC) released a report in 2006–2007 that listed *P. aeruginosa* as the most common hospital-associated bacterial pathogen. In their report of antibiotic-resistant pathogens causing device-associated and procedure-associated, healthcare-associated infections, Hidron et al. [60] placed *P. aeruginosa* at the sixth place. In a hospital setting, the main risk factors for developing severe *P. aeruginosa* infections are medical interventions such as mechanical ventilation [61,62], surgery [63], antibiotic therapy [64,65], and chemotherapy [66,67]. *P. aeruginosa* infections are difficult to treat due to the organism’s inherent and acquired resistance mechanisms to multiple medications [68]. The most common infections associated with *P. aeruginosa* are summarized below.

### 3.1. Burn Wound Infections

Bacterial infections of wounds are common and may sometimes have serious outcomes. The burn wounds become easily contaminated with bacteria, either from external sources, from nearby infected skin, or from internal patient sources [69]. *P. aeruginosa* colonizes wounds from patients’ indigenous flora of the upper respiratory tract and/or gastrointestinal tract in addition to other bacteria and yeasts [70]. Because thermal injury reduces the host defensive peptides produced, opportunistic infections can more easily infect the burn wounds [71]. Up to 75% of burn wound patients die from septicaemia caused by *Pseudomonas* spp. and other bacteria [72]. Sepsis has been identified as a primary cause of deaths in cases of *P. aeruginosa*-infected severe burn wounds [73]. Every year, 265,000 people die from burn injuries according to the World Health Organization (WHO), with about half of those deaths taking place in the South East Asia Region. There are an estimated 7 million burn injuries in India annually, of which 700,000 require hospital admission and 140,000 are fatal [74]. 

### 3.2. Bacterial Keratitis

*P. aeruginosa* causes keratitis in people who wear contact lenses, have had eye surgery, or have ocular diseases. The majority of *P. aeruginosa* infections linked to contact lenses are caused by lens contamination or prolonged lens wear, which damages the cornea’s epithelial surface and increases the risk of corneal abrasions [75]. *P. aeruginosa* induces an opportunistic infection when prolonged contact lens wear compromises epithelial barrier function. *P. aeruginosa* binds to Toll-like receptors (TLR5) on the surface of the cornea, which causes it to be internalized quickly [76]. Blurred vision, photophobia, redness, tears, and abrupt onset and rapid escalation of ocular pain are the hallmarks of *pseudomonas* keratitis. A stromal infiltration and corneal epithelial defect are the clinical outcomes of this illness, which also causes stromal necrosis and gradual thinning [77].

### 3.3. Ear Infection

*P. aeruginosa* is the causative agent of otitis externa, also known as “swimmer’s ear”, which is an inflammation or infection of the external auditory canal. Prolonged exposure to wetness, insertion of foreign items, and *P. aeruginosa*-contaminated water are linked to otitis externa [75]. *P. aeruginosa* is one of the main pathogens that cause chronic suppurative otitis media, also known as chronic otomastoiditis, chronic tympanomastoiditis, and chronic active mucosal otitis media. The bacterium induces long-term inflammation of the mastoid cavity and middle ear, which subsequently leads to tympanic membrane rupture and recurrent ear discharge, also known as otorrhea [78]. 

### 3.4. Infections of the Skin and Soft Tissues

*P. aeruginosa* can cause infections ranging from benign post-surgical and cellulitis-like infections to highly lethal skin and soft tissue infections. It is one of the most frequently isolated pathogens in persistent decubitus ulcers, surgical site infections (SSIs), infections after trauma, and cellulitis in neutropenic individuals. Even while combination of antibacterial and surgical debridement should be the standard of care, certain individuals may still need additional medicinal treatment. While surgical debridement is necessary to remove necrotic tissue from an infected surgical wound or from an infected persistent decubitus ulcer, acute cellulitis typically does not require surgery. Antimicrobial therapy is, without a doubt, crucial in every situation. The best combination of antibiotics often consists of a carbapenem, a fluoroquinolone, or an anti-pseudomonal beta-lactam and is determined by in vitro susceptibility testing. Although antibiotic treatment courses typically last 10–14 days, lesser durations might be an option for patients whose infections are sufficiently controlled, and/or whose clinical signs and symptoms have resolved quickly [79]. The initial symptoms of folliculitis, after extended immersion in contaminated hot tubs, spa pools, whirlpools, and swimming pools, or following leg waxing, are a sudden onset of big, painful, monomorphic papules and pustules. The lesions often develop 8–48 h after exposure and frequently cluster on body regions that come into touch with contaminated water. Folliculitis in immunocompromised persons may develop into ecthyma gangrenosum. *P. aeruginosa* infections in patients with AIDS might result in progressive folliculitis with cellulitis or subcutaneous nodules. “Hot-foot” syndrome is another issue that affects kids. It is distinguished by excruciating plantar nodules [80,81,82].

### 3.5. Necrotizing Fasciitis and Gangrenous Cellulitis

Skin and fascial layer *P. aeruginosa* infections are uncommon but dangerous illnesses. These are characterized by inflammation and damage that occurs quickly and steadily, leading to fulminant skin necrosis and eventual death. Necrotizing fasciitis is an uncommon but dangerous infection of the subcutaneous tissue and fascia that *P. aeruginosa* develops in elderly people with impaired immune systems. Necrotizing fasciitis propagates along the fascial plane in a manner that is directly correlated with the thickness of the subcutaneous layer. Fournier’s gangrene is a particular kind of necrotizing fasciitis; *P. aeruginosa* infection causes scrotal pain and malaise in these individuals, which progresses to blisters, swelling, perineal pain, and necrosis [82].

### 3.6. Green Nail Syndrome/Chromonychia/Fox-Goldman Syndrome

In 1944, Goldman and Fox were the first to report the persistence of pyocyanin in the nail plate and *P. aeruginosa*’s participation; for this reason, the syndrome bears their names [83]. Green nail syndrome is more common in patients with underlying conditions like onycholysis, onychotillomania, chronic paronychia, microtrauma to the proximal nail fold, and associated nail disorders like psoriasis, diabetes mellitus, immunosuppression, and those who are frequently exposed to water or moist conditions [84,85]. Onycholysis and a green-black staining of the nail bed are the condition’s defining features; chronic paronychia is frequently linked to it. The nail plate may be partially or completely involved in green nail syndrome, which is often limited to one or two nails. By touching or scratching their skin, an infected person might spread the bacteria autologously, particularly if the cutaneous surface is damaged [86]. 

### 3.7. Bacteraemia

*P. aeruginosa* bloodstream infection (BSI) is a dangerous condition that needs immediate attention and pertinent clinical judgments to produce a positive result [87]. Hospital-acquired *Pseudomonas* bacteremia was the third most common cause of Gram-negative BSI and accounts for 4% of all cases [88,89]. *P. aeruginosa* was the third most frequent Gram-negative bacterium causing nosocomial BSI, accounting for 4.3% of all cases, according to a statewide surveillance study on nosocomial BSI in the USA [90]. *P. aeruginosa* was shown to be the fifth most frequently implicated isolation in BSI in the intensive care units (ICUs), where it accounted for 4.7% of all cases. In non-ICU wards, it was found to be the seventh most frequently occurring isolate, accounting for 3.8% of cases. The vast majority of crude mortality percentages from extensive monitoring studies that have been published fall between 39% and 48% [82]. Antibiotics are usually used to treat *P. aeruginosa* bacteremia; but, in certain instances, the infection may be hard to cure and may require long-term and/or extensive therapy. In some patient populations, *P. aeruginosa* infection still has significant death rates of up to 62%, despite the advancements in medicine and antibiotic therapy [89]. The BSI can spread to several organ systems, resulting in potentially fatal consequences such as shock, organ failure, and sepsis [91]. Since *P. aeruginosa* is an opportunistic pathogen that prefers immunocompromised individuals, the percentage is much greater in ICU settings [92]. The lack of effective empirical therapy choices at ICUs raises severe concerns about antibiotic resistance in *P. aeruginosa*. Carbapenems, which include imipenem, meropenem, and more recently, doripenem, is the commonly used class of drugs in both empirical and definitive treatment [93]. 

### 3.8. Urinary Tract Infections (UTIs)

Acute, recurring, and chronic urinary tract infections (aUTIs, rUTIs or cUTIs) impact 150 million people annually worldwide. The high rate of recurrent periods and the chronic nature of infections make UTIs the second most frequent bacterial disease after pneumonia, in addition, septicemia is also frequently brought on by UTIs [94]. Antibiotic resistance is continuing to rise, and MDR and XDR uropathogens are starting to arise as a result of the widespread and unchecked use of antibiotics during UTI treatment and prevention [95]. *P. aeruginosa* is a common cause of UTIs, especially in hospital settings and ICUs [96]. It is associated with high mortality rates and often shows resistance to antibiotics. Catheterization and surgery are common causes of *P. aeruginosa* UTIs, and the bacterium is known to form biofilms. The bacterium can cause severe complications such as sepsis and has been found to invade epithelial and mast cells [97].

### 3.9. Respiratory Tract Infections

#### 3.9.1. Cystic Fibrosis (CF) 

The genetic illness CF occurs by a recessive mutation in a gene coding the chloride ion channel CF transmembrane regulator (CFTR). Among the organs and tissues affected by CF are the lungs, liver, gastrointestinal system, pancreas, and male reproductive system, due to the extensive distribution of the CFTR protein channel. The mutation in the CFTR gene causes increased sodium absorption and impaired mucociliary clearance. The unique lung environment in CF patients, along with factors such as stress and antibiotic presence, promote bacterial colonization. This leads to the buildup of mucus in the airways, creating a conducive environment for *P. aeruginosa* colonization. *P. aeruginosa* adapts to this environment by undergoing evolutionary changes. Once it colonizes the lungs, it becomes challenging to eradicate and can be fatal [98,99]. *P. aeruginosa* infections pose a significant risk to individuals with CF, a leading cause of both mortality and morbidity. 

#### 3.9.2. Pneumonia

*P. aeruginosa* can cause nosocomial infections, including pneumonia, and it is the most often found Gram-negative bacterium in pneumonia, particularly in hospital settings [100]. Pneumoniae caused by *P. aeruginosa* fall into four categories: hospital-acquired, ventilator-associated, health care-associated, and community-acquired. *P. aeruginosa* constitutes one of the most frequent etiological agents of ventilator-associated pneumonia (VAP) in the United States and Europe. VAP caused by *P. aeruginosa* presents special difficulties for therapeutic care. The risk factors for *P. aeruginosa* VAP are old age, prolonged mechanical ventilation, a history of *P. aeruginosa* colonization, and previous antibiotic treatment, as well as admission to a ward where *P. aeruginosa* infections are common, solid malignancy, and shock [101,102,103].

#### 3.9.3. Bronchiectasis

Bronchiectasis is the term used to describe chronic bronchial dilatation. Consequently, there is insufficient mucus drainage and a higher chance of bacterial infection. Because *P. aeruginosa* is an opportunistic bacterium, patients with CF, other types of bronchiectasis, and severe chronic obstructive pulmonary disease are particularly susceptible to infection. When *P. aeruginosa* becomes a chronic infection in bronchiectasis, it is seldom eradicated even with intensive intravenous antibiotic therapy. Extended lung damage and more severe airflow restriction are associated with persistent infection [104].

### 3.10. Joint and Bone Infections

The most prevalent cause of revision following total knee arthroplasty and the third most common cause after total hip arthroplasty is periprosthetic joint infection (PJI) [105,106,107], despite the fact that total hip and knee arthroplasties have been shown to be effective in improving patients’ function and quality of life over the long term. PJI, which can happen in up to 2% of first surgeries, is regarded as one of the most severe complications following total joint arthroplasty [107]. Although bones and joints are normally sterile regions, germs can enter them by exogenous and endogenous contiguous foci of infection or hematogenous dissemination [108]. *P. aeruginosa* was found in 4.4% of cases of 414 patients with osteomyelitis [109]. *P. aeruginosa* was implicated in 10% of all cases of sternoclavicular septic arthritis, with common risk factors including intravenous drug use, diabetes mellitus, trauma, and infected central venous lines [110].

### 3.11. Chronic Rhinosinusitis and Otitis Media 

An inflammation that lasts for 12 weeks or more in the paranasal sinuses and nose is called chronic rhinosinusitis (CRS). Bacterial biofilms boost bacteria’s resistance to antibiotics in different ways and have been linked to recalcitrant CRS. These include anionic charges from extracellular DNA inside the biofilm matrix, gene expression variances, enzymatic deactivation and bacterial metabolic variability within the biofilm [111]. In one study, up to 54% of CRS patients had biofilms on their sinonasal mucosa, compared to 8% of control patients [111]. Furthermore, numerous studies have found that patients undergoing revision surgery had a greater prevalence of biofilm [111,112]. Following endoscopic sinus surgery, symptoms and indicators of CRS have been linked, in particular, to the presence of biofilm-forming *P. aeruginosa* strains [113]. According to disease severity measures like the Sinonasal Outcome Test-22 (SNOT-22) and Visual Analogue Scale (VAS), 9% of CRS patients have *P. aeruginosa* in their sinuses, and it has been associated with a worse quality of life [113]. CF patients frequently have sinus infections caused by *P. aeruginosa*, with the species being found in sinus cultures of as many as 49 percent of CF patients with CRS [41]. 

### 3.12. Biofilm

Biofilms are aggregated clumps of bacterial cells that produce the elements of the extracellular matrix necessary to keep the community intact. The biofilm mode of development protects cells from different chemical and environmental threats, such as phagocyte engulfment, while allowing cells to stay near resources and encouraging genetic material interchange [114]. Bacteria that form biofilms usually exhibit several phenotypic differences when compared to the identical strains of bacteria grown in planktonic culture. Changes in motility, irregular increase in the extracellular polysaccharide synthesis, and enhanced antibiotic resistance are among these alterations [115,116]. After five days of biofilm growth, variations in the expression of over 70 *P. aeruginosa* genes were observed using DNA microarray analysis [117]. *P. aeruginosa* biofilm formation takes place in five stages, each being different from planktonic bacteria both in their visible phenotype and unique protein patterns, and the formation cycle appears to be intricately controlled as evidenced by the larger than anticipated changes in protein patterns between the stages [118]. Given all of these variables, treating infections caused by MDR strains poses additional challenges, making it practically hard to eliminate *P. aeruginosa* from infections with biofilm formation, such as those that cause CF [119]. The morbidity and mortality of patients are impacted by these problems. Other effects include increased frequency and length of hospital stays, as well as higher treatment expenses [120,121]. When coming into contact with biofilms, macrophages acquire the ability to secrete substances and transform into tissue-damaging cells [122]. *P. aeruginosa* infections constitute a severe hazard as it is difficult to employ common antimicrobials for therapeutic care in the clinical setting. Antimicrobial peptides, quorum sensing (QS) inhibitors, biofilm-degrading enzymes, and iron scavengers are a few antimicrobial remedies that have been used in the attempt to target iron metabolism, promote biofilm dispersal, and prevent QS [123].

## 4. Phage Therapy in Humans—Progress and Limitations

Phage therapy has been used to treat a variety of bacterial infections, including *P. aeruginosa* infections [124]. However, rather than extensive randomized controlled trials, most of the clinical experience with phage therapy on humans has been reported as small case series or case reports. Finding the right phages for each unique bacterial infection is one of the difficulties in carrying out clinical trials of phage therapy. This can be a time-consuming procedure that calls for the creation of special phage formulations for every patient. Despite these difficulties, phage therapy for *P. aeruginosa* infections in people has been the focus of multiple clinical trials. To infer the possible advantages of phage treatment from the published research is difficult, for several reasons [125].

Many clinical articles lack a control group.There is variation in the length of phage therapy used among studies, as well as in the mode of administration, phage dose, or the number of different phages (such as monotherapy vs. cocktail).It is challenging to pinpoint the specific roles of phages in affecting clinical outcomes since they are frequently provided in combination with antibiotics.Phage susceptibility testing (PST) has not been a consistent and standard procedure for confirming whether prescription phages are successful against a bacterial disease.Furthermore, publications with negative results are less likely to be published than ones with positive results due to publication bias.

A list of the most common human *P. aeruginosa* infections treated with phages is presented in Table 2. In a classical clinical trial study, chronic otitis media caused by *P. aeruginosa* was treated with a six-phage cocktail [126]. The results showed that *P. aeruginosa* counts in the phage therapy group decreased significantly with no detectable local or systemic side effects, and that the bacterial counts remained rather stable in the placebo group. The same phage cocktail was later used to treat dogs suffering from *P. aeruginosa*-induced chronic otitis media, and the results were promising as both the clinical scores and bacterial counts decreased in phage-treated ears [127]. Patients who have serious conditions, multiple underlying illnesses, or long-term indwelling urinary catheters are at a higher risk of developing complicated or resistant bacterial infections [128]. There are anecdotal, undocumented examples of patients with recurrent UTIs who have benefited from phage therapy combined with antibiotics. The uropathogens residing in the microbiomes of the gut, vagina, and urine, presumably play a role in the pathogenesis of recurrent UTIs [129,130]. *P. aeruginosa* biofilms of urinary catheters play an important role in the pathophysiology of UTIs. Catheters coated with the benign *E. coli* strain HU2117 and lytic *Pseudomonas* phage φE2005-A prevented the development of the biofilms [131]. Mittal et al. showed that otopathogenic *P. aeruginosa* can penetrate and survive inside macrophages by the use of assays based on human and animal cells [82,132]. 

## 5. Immune System and Phage Therapy

The liver and spleen include components of the mononuclear phagocyte system, which removes foreign particles from the bloodstream, including phages. Since phage titers are often greatest in the spleen and liver, these organs have been recognized as the primary locations of phage accumulation [142]. A significant portion of professional phagocytes are present in both organs. Immune cells in the liver and spleen appear to be the primary mechanism in the human body for neutralizing phages by phagocytosis [143,144,145,146,147,148]. It should be noted that phagocytosis allows for the elimination of phage particles even in the absence of a developed defense against phages. As a result, the majority of animal or human cells that interact with phages in vivo are most likely phagocytes. Tiwari et al. [149] were the first to present conclusive data on phage cooperation with the innate immune system. They demonstrated that neutrophil–phage cooperation is essential for the clearance of *P. aeruginosa* infections. On the other hand, when a single phage is utilized during phage therapy, the presence of neutrophils is a prerequisite for the elimination of the emerging phage-resistant bacteria [150]. The results were confirmed by Roach et al. [151] and Pincus et al. [152] and converted into an in silico model by Leung and Weitz [153]. Recent research on the cellular immune response triggered by phages has been carried out both in vivo [154,155] and in vitro [156,157,158]. These studies have demonstrated the ability of phages to interact with the human immune system. It is important to remember that much research [155,157] on the immune response triggered by phages has been conducted with lysates of phages that are contaminated with bits of the host bacterial cell wall stuck to the phage tails or remnants of lysed bacteria (like LPS, cytosolic proteins, or membrane particles). Because of this, it is quite challenging to identify the specific phage elements that really modulate the immunological response. The effect of host immunity on phage treatment efficacy in treating acute pneumonia induced by MDR *P. aeruginosa* was investigated in a mouse model [151]. Studies comparing the effectiveness of phage therapy and preventive therapies in healthy lymphocyte-deficient, MyD88-deficient and neutrophil-deficient mice showed that the synergy of neutrophils is necessary to cure pneumonia. The monophage treatment was imitated by the phage PAK_P1. The results demonstrated that phage resistance could arise in immunologically sound hosts but did not always lead to treatment failure. Instead, if the innate immune system of the host was unable to eradicate phage-resistant subpopulations, it could lead to a treatment failure [151].

The human immune system can respond to pathogenic microorganisms in multiple ways. As a first line of defense, neutrophils can eliminate pathogens through phagocytosis, render them inactive by deploying neutrophil extracellular traps (NETs) or reactive oxygen species (ROS). Neutrophil responses to phages remain mysterious despite recent research demonstrating their substantial proportionality in human microbiomes and their current exploration as antibacterial treatments. In a study using the lytic *P. aeruginosa* phage PAK_P1, a 93 kb myovirus of genus *Pakpunavirus*, a wide range of responses of both resting and stimulated neutrophils from human peripheral blood were determined. No indications that phage PAK_P1 had an impact on phagocytosis or oxidative burst were detected when the neutrophil/phage ratio was increased up to 1:10,000, or NET-osis. A modest rise in IL-8 at the maximum neutrophil/phage ratio was the only significant signal seen. The study’s findings demonstrated that it is unusual for phages to inadvertently trigger excessive neutrophilic responses, which can harm tissues and exacerbate illness. Phage-stimulated IL-8 synthesis has the potential to regulate certain host immunological responses since it functions as a chemoattractant and guides immune cells to areas of infection and inflammation [159].

## 6. Phage Formulations Used in Treatments

Phage formulations can be administered on their own or in conjunction with probiotics, synbiotics, or antibiotics [160]. Now, when drug-resistant bacteria are on the rise, it is preferrable to use fewer antibiotics and instead focus on novel, alternative therapies like phage therapy. Furthermore, phage-antibiotic synergy (PAS), i.e., when the combined effect on bacterial elimination is more than the sum of each ingredient alone, has been reported to provide significantly better outcomes in the fight against bacterial illnesses [34,161,162,163,164]. Certain phages that are utilized alone or in conjunction with probiotics as vectors for nutrient production or degradation that may benefit the host can be genetically modified thanks to the development of genetic tools. A Russian laboratory created a mix of six different phages to solve this issue [165]. There are several approaches to phage therapy [166,167,168,169]; however, because multiple phage types may infect the same species or strain of bacteria, phage cocktails should be utilized in phage treatments to combat the quickly occurring phage resistance. 

To achieve acceptable physical biostability of the phage formulation (gel, suspension, solution, or powder), a twin challenge is offered to the phage formulation process. Similar to some proteins, phages may face stability problems in solutions. Phages can be thought of as giant protein complexes that encapsulate genetic material (DNA or RNA). Therefore, techniques for protein stabilization have been used in the development of phage treatment products [170]. The preferred technique for delivering phage to the lung is nebulization. Phages can be supplied as inhalable aerosol droplets with little to no titer loss, depending on the sort of nebulizer being employed [171,172]. Nebulized phage aerosols have been utilized for patients with respiratory infections that antibiotic treatments failed to cure [173]. An advanced model of airway surface fluid infection that replicated the environments of healthy lungs of CF patients in vitro showed that the jumbo phage PA5oct significantly reduced the planktonic and resident *P. aeruginosa* populations [174]. The benefits of dry powder formulation include simplicity in administration, storage, and transportation. Dry powder inhalers (DPIs) are smaller, more portable, and require less electricity to function, resulting in faster treatment times than nebulizers. Furthermore, DPIs do not require routine disinfection and cleaning [175]. Dry powders that can be stored at room temperature do not need cold chains and have decreased costs [176]. The efficiency of leucine (45–10%) and lactose (55–90%) in eight combinations as excipients was studied for the shelf-life of spray-dried phage powders using three *P. aeruginosa*-specific PEV phages (PEV1, PEV20, and PEV61) [177]. These two excipients, when packaged in dry conditions and stored for a year at 20 °C and 60% relative humidity, gave both biological and physical stability for the phages. Specifically, higher lactose concentration had a significant impact on phage viability offering superior phage protection. The phage powders showed no toxicity to macrophages and epithelial cells [177]

### 6.1. Phage and Antibiotic Synergism (PAS)

The presence of some antibiotics stimulate the bacterial host to produce more phages and to produce larger plaques [178,179]. In addition, when administered at sublethal concentrations, several antibiotics improve the productivity of bacteriophages from bacterial cells [179,180]. In the case of PAS, smaller amounts of antibiotics are needed for the treatments, and over time, the risk of selection of bacterial antibiotic resistance is reduced [181,182]. Combining phage with antibiotics reduces the probability of resistance development, raises phage and/or antibiotic absorption into biofilms, and supports the elimination of the bacterial pathogen [183]. Phages may even act as selective agents for spontaneous mutations in multi-drug resistant bacteria that are more susceptible to antibiotics through an evolutionary trade-off [184]. For instance, as the ϕKZ phage encodes its own RNA polymerase (RNAP), and does not require host RNAP for transcription, combining the phage with an antibiotic such as rifampin that block the bacterial RNAP will be advantageous. Also, other antibiotics that block bacterial processes not essential for phage propagation can also be used in combination with phages [52,185]. Ceftazidime, ciprofloxacin, piperacillin, and meropenem are drugs that work together with *Caudoviricetes* phages when administered at different dosages [34,183,186]. 

Phage-antibiotic interactions are not always beneficial; sometimes, they can be adverse or just neutral [34,182,183]. Given that antibiotics and phages have different modes of bacterial killing, it has been postulated that the combination of the two agents may be more successful at managing bacteria than each one acting alone [178]. Additionally, Chaudhry et al. describe how phages and antibiotics interact within *P. aeruginosa* biofilms, observing that phages and high levels of tobramycin are antagonistic to each other. High tobramycin concentrations (8× minimal inhibitory concentration, MIC) reduced biofilm density more than low ones (1× MIC) did before being combined with phages. When combined with phages, the high level of tobramycin plus bacteriophage treatment proved to be less effective than the first 8× MIC treatment. Phage combination therapy with 1× MIC tobramycin proved to be substantially more effective. Tobramycin’s suppression of phage replication at high concentrations or the antibiotic’s ability to lower bacterial cell density to a threshold where bacteriophages have difficulty replicating were proposed as explanations for this phenomenon [34]. This may be related to the observation that the replication of bacteriophages depends on the density of surrounding host bacteria. The bacteria need to reach a proliferation threshold that is the minimum bacterial concentration needed for the phages to find a new host for replication [187]. If antibiotics lowered the number of bacteria below the threshold before administering the phage, phages would likely be useless as they would not be able to propagate across the bacterial population. 

In a PAS study, the impact of phage PEV20 with five different antibiotics against three *P. aeruginosa* strains isolated from the sputum of CF patients was carried out [188]. Time-kill tests were used to evaluate bacterial killing to determine the synergism between the antibiotic and the phage. The strongest PAS effect was demonstrated by PEV20 and ciprofloxacin when compared to other phage-antibiotic combinations. Two combinations of phage-ciprofloxacin were created using vibrating mesh aerosol and air jet nebulizers. It was also shown that the synergistic antibacterial action tolerated well the nebulization. Phage PEV20 and nebulized ciprofloxacin together showed encouraging antibacterial and aerosol characteristics for 70% of the results [188]. Antibiotics and phages were also shown to work together to reduce the bacterial population in wound infections caused by MDR *P. aeruginosa* [189]. In a study employing a model of chronic lung infection in neutropenic mice, the in vivo efficacy of an inhalable powder of the *Pseudomonas* phage PEV20 produced by spray drying with ciprofloxacin was demonstrated [190]. When ciprofloxacin and PEV20 were administered together, the number of bacteria in the lungs was greatly decreased; however, neither medication alone was able to do so. The synergistic basis for the action is evidenced by the fact that the lethal effect of the combined powder was substantially larger than the additive effect of the separate treatments, both of which had no effect after 24 h. 

### 6.2. Hydrogels and Bacteriophages

Biologics, such as phages, have been administered or delivered under control to target sites, such as wounds and implants, using hydrogels as a carrier [191,192,193,194,195,196]. Non-toxic polymeric materials called hydrogels have three-dimensional networks and are hydrophilic. They are crucial for having a high-water content because they create a biocompatible environment that is ideal for phages as living molecules [196,197]. In addition, hydrogels simulate biological tissues by retaining a significant amount of water in their matrix. This creates the ideal conditions for proteins, live cells, and other biomolecules to be accommodated, hence broadening their use in the biomedical area [198,199]. Furthermore, the hydrogel system’s programmable physical characteristics and biodegradability allow for regulated drug release, a feature that is equally relevant to the transport of biomolecules [200,201].

Given their favorable properties for biological agent incorporation, hydrogels show great promise as a medium for phage delivery. Phage hydrogels have been used to treat or prevent MDR bacterial infections thus exploiting the advantages of both phages and hydrogels. An increasing body of preclinical in vivo and in vitro research suggests that hydrogels could be the perfect phage delivery vehicle. Hydrogels containing ciprofloxacin, phages, or phages plus ciprofloxacin, as well as hydrogels without any additives, were produced in a study to compare their antimicrobial effects [51]. A mouse wound infection model was used to study the antibacterial activities of these hydrogels both in vivo and in vitro. The healing process of wounds in several mouse groups demonstrated that hydrogels containing phages and hydrogels containing antibiotics have almost identical antibacterial effects. However, phage-containing hydrogels performed better than antibiotics alone in terms of pathological process and wound healing, and the best performance was obtained with the phage-antibiotic hydrogel, demonstrating a clear PAS effect of the phage cocktail and the antibiotic. 

### 6.3. Phage Cocktails

A “cocktail” of many phages can be used for phage treatment, or it can be performed with a single phage (monophage). For the monophage treatment, a phage with the broadest host range should be chosen [202,203]. To broaden the host spectrum, phage cocktails may be tailored by combining different isolates. If resistance develops, the cocktails can be later reconstituted [204,205,206,207]. The bacterial strains that become resistant to one phage can be targeted by other virulent phages added to the cocktail, in contrast to monophage treatment where the spectrum of bacteriolytic activity is limited [208].

Phage cocktails have been used in response to many inadequate-to-moderate results seen during the evaluation of monophage preparations [209]. Several strategies have been studied to increase the usefulness of both cocktail and monophage preparations [210]. Only a few studies using phage cocktails to target *P. aeruginosa* infections have been published [210]. A cocktail consisting of two previously characterized *Pseudomonas* phages (PAK_P1 and PAK_P4) and of four novel phages (DEV, PYO2, E217, and E215) was constructed and used to cure *P. aeruginosa* bacteremia in wax moth (*Galleria mellonella*) larvae infection model and to treat acute respiratory infections in mice [42]. Treatment of wax moth larvae with a phage cocktail before bacterial infection demonstrated its preventive effect. Overall, the study showed that the phage cocktail was successful in clearing biofilms from experimentally infected animals and accelerating their response to therapy [42].

## 7. Phage Therapy Treatments of *P. aeruginosa* Infections

A summary of the effectiveness of phage therapy in treating *P. aeruginosa* infections is presented in Table 3.

### 7.1. Ventilator-Associated Pneumonia (VAP) 

According to recent evidence, the isolation of MDR bacteria takes place frequently from samples of VAP cases caused by *P. aeruginosa*. As compared to non-MDR infections, MDR *P. aeruginosa*-related pneumonia appears to be a significant factor in determining excessive ICU stays, prolonged mechanical ventilation, and increase in-hospital mortality.

A study conducted on a porcine infection model demonstrated that the choice of an active phage cocktail is important. In this case, the active phage cocktail, supplied by Pherecydes Pharma, contained five lytic *Pseudomonas* phages of three myo- and two podoviruses. The optimized aerosol delivery conditions allowed for the delivery of high phage concentrations in the lungs resulting in rapid reduction in *P. aeruginosa* numbers in the lungs [211]. 

### 7.2. Urinary Tract Infections (UTIs)

To treat UTIs, a variety of phage products has been used, such as lytic phages isolated from the environment, phage cocktails, lytic phage enzymes, genetically engineered or modified phages, and suitable synergistic phage-antibiotic combinations [212,213,214,215,216]. The aqueous environment of the urinary tract makes it an ideal location for medicinal applications. Phage movement is severely restricted in a low-moisture environment, which also indirectly reduces their lytic efficiency. The liquid environment, on the other hand, increases the possibility of phage particles making contact with their bacterial hosts by promoting multidirectional mobility of the particles [217]. The urinary bladder is easily accessible via catheterization from an anatomical perspective [218], and urinating or irrigation can successfully eradicate any residual germs, at least in some situations. Phages appear to tolerate urine to different degrees. In a study by Cardoso et al. [219], mice were used to evaluate the biological effects of the radiolabeled *P. aeruginosa* phage PP7 following intravenous injection. Regardless of the general state (normal mice vs. infected mice), phages were primarily discovered in the urinary bladder within 3 h. Phage buildup in the bladder surpassed 50% of the administered dose in all groups. 

### 7.3. Respiratory Infections 

The preferred technique for delivering phages to the lung is nebulization. Phages can be supplied as inhalable aerosol droplets with little to no titer loss, depending on the sort of nebulizer being employed [171,172]. Nebulized phage aerosols have been utilized for patients with respiratory infections that failed to cure with antibiotic treatments [173]. Chronic airway infection, primarily from *P. aeruginosa*, is one of the main problems in CF patients, and to manage the infection, CF patients require recurrent antibiotic treatments. Cafora et al. investigated the use of phage therapy in a CF background for treating *P. aeruginosa* infections. They used zebrafish as a model system and observed the efficacy of phage treatment over time. The results showed a significant reduction in embryo lethality and a drop in cytokine expression in CF embryos treated with the phage cocktail [46]. 

### 7.4. Periprosthetic Joint Infections (PJI) 

Patients with relapsing PJI, primarily the elderly, may not be candidates for repeated surgeries, particularly in the knee joint, due to severe bone loss and infection that could make revision surgery more challenging or even necessitate amputation. The existence of a dense biofilm, which prevents standard antimicrobial therapy from completely eliminating the pathogens [68,220], or the development of antimicrobial resistance [221,222], may be to blame for the partial success of these surgical operations. The re-evaluation of phage therapy as an adjunct to conventional antibiotic therapy was prompted by the search for new management options [223]. In several case studies of PJI caused by *Staphylococcus epidermidis* or *P. aeruginosa*, local injection of phages with implant retention and debridement were used in patients. These cases are presented in Table 2.

### 7.5. Cardiac Device-Associated Infections

Heart infections brought on by prosthetic valves, ventricular assist devices, vascular grafts, and cardiovascular implanted electronic devices have all been treated with phage therapy (Table 2). A patient with *P. aeruginosa* had a seven-hour IV phage infusion. Over the next five days, the phages were applied locally during surgical debridement and once again through an indwelling drain every twelve hours with promising results [48]. 

### 7.6. Wound Infections

There is a wealth of information demonstrating that phages help treat wound-associated infections [224]. The effect of a three-phage cocktail was studied on 20 patients with non-healing chronic wounds that did not respond to antibiotic therapy and conventional local debridement [225]. The patients were between 12 and 60 years of age, and five had the wound infected with *S. aureus*, six with *E. coli*, and nine with *P. aeruginosa*. To eradicate pathogens from the wound surface, the specially designed phage cocktail was administered topically to the wound on different days. Average bacterial counts were clinically evaluated at the time of visit and after three and five phage doses were administered and compared. The total number of leukocytes was significantly reduced according to various blood parameters when compared between two times of bacteriophage treatment [225]. Rezk et al. evaluated the therapeutic efficacy of phage ZCPA1 in rats for the treatment of full thickness wounds infected with *P. aeruginosa* strain. Wound closure percentage, bacterial count and histopathological analysis within 17 days showed that phage ZCPA1 can be a promising antibacterial agent [226].

The primary complications of severe burns are bacterial infections, which can cause sepsis, develop numerous organ failures, and even postpone wound healing. Infections are also a significant contributor to burn-related fatalities. Phage therapy is frequently used in the treatment of burn wounds; it has, in numerous instances, demonstrated therapeutic potential. Oral administration combined with a topical application of a phage cocktail was used in Wroclaw, Poland, to treat burn patients whose wounds were infected with *P. aeruginosa*, *S. aureus*, *K. pneumoniae*, or *Proteus*. Pathogens were not present in samples from phage-cocktail-treated skin of 85% (42/49) of the patients, and in the remaining patients, the clinical symptoms were dramatically alleviated despite the presence of the pathogens in the burn wounds [227]. Skin infections permit topical phage delivery, just like rhinosinusitis does [228].

By contrast, phage therapy reduced the bacterial bioburden in *P. aeruginosa*-infected burn wounds more slowly than standard care did in the clinical trial “PhagoBurn”, involving 13 patients randomly assigned to phage therapy alone and 14 patients who were randomly assigned to standard of care alone. This study was conducted in nine burn centers in Belgium and France. Patients with a burn site infected with *P. aeruginosa*, and who were 18 years or older, were eligible for the study. The patients were randomly assigned to receive either a cocktail of 12 naturally occurring lytic anti-*P. aeruginosa* bacteriophages or the standard of treatment 1% sulfadiazine silver emulsion cream. The goal was to determine the median amount of time needed to reduce consistently the bacterial load by at least two quadrants. The study found that the phage treatment took a median of 144 h to achieve the goal, whereas the standard of care group took 47 h. However, the phage cocktail’s poor stability caused the participants to receive less of the phage dosage than planned, rendering the treatment ineffective. The study revealed that modest phage dosages were ineffective against the bacteria recovered from individuals whose phage treatments had failed [229].

### 7.7. Biofilms

Phages allow for another approach to control *P. aeruginosa* biofilms. The *P. aeruginosa*-specific phage vB_PaeM_P6, with a sub-MIC of ciprofloxacin, prevented biofilm formation for 95% of the tested clinical strains when ciprofloxacin alone resulted in only 20% biofilm inhibition, and phage treatment alone resulted in 50% biofilm inhibition [230]. Phages can pierce biofilm structures and lyse biofilm cells in addition to being successful at eliminating planktonic bacteria. For example, two lytic *P. aeruginosa* phages, vB_PaeP_MAG4 (MAG4) and vB_PaeM_MAG1 (MAG1), dramatically decreased the biofilm with MAG4 having a greater immediate efficiency [231]. Similarly, the myovirus phages, vB_PaeM_SCUT-S1 and vB_PaeM_SCUT-S2, were effective in removing existing biofilms and inhibited the growth of *P. aeruginosa* strain PAO1 at low infection levels [232]. A lytic bacteriophage vB_PaeM_LS1 demonstrated the potential to prevent and disperse the *P. aeruginosa* biofilm in static settings for 48 h [233]. A MDR *P. aeruginosa* strain was eliminated by the phage MA-1 both as a biofilm and as planktonic cells [234]. *P. aeruginosa* isolates of 44 different mutilocus sequencing types from both CF and non-CF individuals with CRS from three continents were used to grow in vitro biofilms, and the anti-biofilm activity of a four-phage cocktail was tested. It was found out that the biofilm biomass was reduced by 76% in 48 h on average [41].

### 7.8. Keratitis, and Oral and Eye Infections

Oral infections caused by *P. aeruginosa* can lead to mouth ulcers that present a challenge for oral disease specialists. Phages belonging to the *Plasmaviridae* and *Inoviridae* families were isolated from enhanced sludge and showed efficacy in the management of carbapenem-resistant isolates. The phages may be utilized in conjunction with other therapies like antibiotics or mouthwashes, particularly in severely infected regions like oral ulcers [235]. A study was carried out to examine the efficacy of phage treatment of bacterial keratitis. The results showed that the phages may prevent *P. aeruginosa*-induced keratitis in a mouse model of keratitis and that the phage may be a more efficient preventive than the antibiotic treatment for horse keratitis [236]. 

### 7.9. Adjustment of the Microbiota Diversity and Composition

Bacteriophages that exist naturally in microbiomes are crucial in preserving equilibrium of the bacterial population [237,238,239,240,241,242,243,244,245,246,247,248]. Novel methodologies to study gut microbiome interactions have been used to identify prolonged interactions between the microbiome components (phages, bacteria, viruses and other micro-organisms) that might be associated either to disequilibrium or healthy equilibrium [249]. The use of phages to alter the bacterial population of the microbiota is one of the most intriguing approaches on therapeutic modulation [239,250]. Phages can be used as compassionate therapy in acute instances or to rebalance the microbiota in chronic disorders. The benefit of using phage cocktails in phage therapy to lower the possibility of resistance is well acknowledged in both situations. Research conducted in vivo on the interplay between gut microbiota and bacteriophages has changed our knowledge of the replication, survival, and life cycle of phages [249,251]. 

Low bacterial diversity in an unbalanced microbiota may lead to a higher percentage of pathogenic bacteria that promote mucosal inflammation. Currently utilized methods of microbiota manipulation include fecal microbiota transplantation (FMT), probiotics, and certain diets. As an alternative, harmful bacteria might be reduced specifically by the use of lytic phages [252]. The population of bacteria can be positively modulated using phage therapy. When these phage mixtures were examined and evaluated for toxicity and unfavorable effects, no side effects were noted. Phage training for clinical purposes is beneficial for two reasons. Firstly, it increases the success rate of phage therapy by having phages for all clones within a given strain. Secondly, it simplifies the production process by having access to a limited number of phages or even a single phage covering almost all circulating strains of a pathogen [166,252].

**Table 3 viruses-16-01051-t003:** Evaluating the effectiveness of some *Pseudomonas* bacteriophages in vitro and in vivo.

Phage	Disease/Type of Infection	Application	Outcome	Ref.
KPP10	Pneumonia and sepsis	Respiratory	Decreased levels of TNF α, IFN-γ and IL-1β in the serum and bacterial count in the serum and lung	[58]
PEV20	Lung infection	Respiratory	Showed the potential of pulmonary administration of PEV20 phage in a dry powder formula to treat lung infections caused by MDR *P. aeruginosa*	[253]
MYY9 HX1 MYY16 TH15	Acute pneumonia, chronic pneumonia	Respiratory	Reduced the symptoms of infection and disease progression, inflammatory responses, and lung damage in mice	[26]
KTN4	Biofilms and infections	Systemic	Gentamicin exclusion experiment on NuLi-1 and CuFi-1 cell lines significantly inhibited the internalization of wild-type *Pseudomonas* into CF epithelial cells, resulting in a 4–7 log reduction in extracellular bacterial burden. It also decreased the synthesis of pyocyanin and siderophore	[53]
E215 DEV PYO2 E217 PAK_P4 PAK_P1	CF	Respiratory	Successful treatment of bacteremia (*Galleria mellonella*) and acute respiratory infection of mice	[42]
ΦPan70	Biological activity in the model of burnt mice	Systemic	ΦPan70 reduced the biofilm formation and existing biofilms. ΦPan70 reduced the number of planktonic cells	[54]
vB_Pae_SMP5 vB_Pae_SMP1	Thermal injury	In vitro and vivo	Hydrogel-shaped bi-phage mixture that effectively reduced CRPA infections in wounds and accelerated burn wound healing	[37]
PB10 PA19	Wound infections (Antimicrobial effect)	In vitro and vivo	Phage cocktail hydrogel was effective in reducing the bacterial load of infected wounds	[51]
DSM 19872 (JG005) DSM 22045 (JG024)	Pneumonia	Systemic	Induced a minimal humoral response in the form of phage-specific antibodies	[49]
Pa193 Pa204 Pa222 Pa223	Chronic rhinosinusitis (CRS)	Systemic	Reduced the biomass of *P. aeruginosa* biofilm in sinuses infected with CT-PA phage cocktail	[254]
PP1450 PP1777 PP1902 PP1792 PP1797	Pneumonia	Respiratory	Inhaled bacteriophages were effective in reducing the bacterial load of pigs infected with PA	[211]
DMS3 and PEV2	CF	In vitro	PEV2 and DMS3 were both able to stop the growth of bacteria in PAO1-NP and PAO1-WT infection models, respectively	[255]
vB PaeP-SaPL	Bacteremia	Systemic	Was effective at inhibiting the growth of bacteria, selectively against the majority of *P. aeruginosa* strains, and capable of boosting mouse survival	[43]
RLP	Bacteremia	Systemic	When compared to the untreated group, RLP-treatment of bacteremic mice infected with *P. aeruginosa* strain PA-1 showed a survival rate of 92%	[31]
Bϕ-R656 Bϕ-R1836	Pneumonia	Systemic	Increased the survival of *Galleria mellonella* larvae at 72 h after infection and pneumonia model mice at 12 days after infection	[38]
MPs	Pneumonia or CF	Respiratory	Prevented mortality from pneumonia	[256]
BrSPI	Culture	In vitro	Effective in suppressing bacterial growth up to 12 h after infection	[50]

## 8. Conclusions

The absence of effective antibiotic medicines has led to the fast rise and spread of drug-resistant and MDR bacterial strains, posing a severe threat to public health. Due to its opportunistic nature, *P. aeruginosa* can fatally infect persons with CF, with severe burns, and in other vulnerable states, resulting in severe infections. Sadly, MDR *P. aeruginosa* cannot be eradicated by conventional antibiotics due to a variety of antibiotic resistance mechanisms, including acquired and innate drug resistance, as well as its capacity to form biofilms. To combat these superbugs, there is an urgent need for new antibacterial treatments. Phages have been utilized as antibacterial agents for almost a century, but recently, there has been an increased interest in them because of their high specificity and abundance. According to the investigations, the results indicate the effectiveness of *P. aeruginosa* bacteriophages in the form of phage cocktails, or as monophages, in some cases combined with sub-inhibitory concentrations of antibiotics. However, there is still a lack of research on phage treatments of human patients.

## Figures and Tables

**Table 1 viruses-16-01051-t001:** Characteristics of *P. aeruginosa* bacteriophages used for therapeutic purposes. All phages have dsDNA genomes except for the cystovirus phage phiYY with dsRNA genome whose taxonomic (realm, kingdom, phylum, class) position before the family level is *Riboviria* > *Orthornaviriae* > *Duplornaviricota* > *Vidaverviricetes* while that of the others is *Duplodnaviria* > *Heungongisvirae* > *Uroviricota* > *Caudoviricetes*.

Family or Subfamily	Genus	Species	*Pseudomonas* Phage Name	Phage Therapy Use	Genome Size (bp)	Ref.
*Autographiviridae*	*Phikmvvirus*	*Phikmvvirus LUZ19*	LUZ19	In vitro	43,548	[25]
		*Phikmvvirus HX1*	HX1	In vitro and vivo	43,113	[26]
		*Phikmvvirus PAXYB1*	MYY9	In vitro and vivo	43,337	[26]
		*Phikmvvirus PNM*	PNM	In vitro and Human case study	42,721	[27,28]
		*Phikmvvirus MPK6*	MPK6	In vitro and vivo	42,957	[29]
		*Phikmvvirus MPK7*	MPK7	In vitro and vivo	42,874	[30]
		*Phikmvvirus phiKMV*	phiKMV	In vitro	42,519	[30]
		*Phikmvvirus RLP*	RLP	In vitro and vivo	43.446	[31]
	*Stubburvirus*	*Stubburvirus LKA1*	LKA1	In vitro	41,593	[32]
*Queuovirinae*	*Nipunavirus*	*Nipunavirus JG054*	JG054	In vitro	57,839	[33]
		*Nipunavirus NP1*	NP1	In vitro	58,566	[34]
		*Nipunavirus PAJD1*	vB_PaeS_PAJD-1	In vitro and vivo	57,919	[35]
		*Nipunavirus PaMx25*	PaMx25	In vitro and vivo	57,899	[36]
-	-	*Septimatrevirus vB_Pae_SMP5*	vB_Pae_SMP5	In vitro, human case study	43,070	[37]
		-	vB_Pae_SMP1	In vitro, human case study	Not reported	[37]
-	*Casadabanvirus*	*Casadabanvirus Bϕ-R1836*	Bϕ-R1836	In vitro and vivo	37,714	[38]
		*Casadabanvirus DMS3*	DMS3	In vitro	36,415	[39]
*Schitoviridae*	*Litunavirus*	*Litunavirus Ab09*	Ab09	In vitro	72,028	[40]
-	*Bruynoghevirus*	Uncl. *Bruynoghevirus*	Pa223	Human case study	45,703	[41]
		Uncl. *Bruynoghevirus*	Pa222	Human case study	45,770	[41]
		*Pbunavirus MYY16*	MYY16	In vitro	Not reported	[26]
		*Pbunavirus JG024*	JG024	In vitro	66,275	[33]
		*Pbunavirus PYO2*	PYO2	In vitro	Not reported	[42]
		*Pbunavirus vB PaeP-SaPL*	vB PaeP-SaPL	In vitro and vivo	45,796	[43]
		*Pbunavirus PT07*	PT07	Human case study	94,660	[44]
		*Pbunavirus DL68*	DL68	In vitro	66,111	[45]
		*Pbunavirus Pa193*	Pa193	In vivo, human case study	66,657	[41]
		*Pbunavirus E217*	vB_PaeM_E217	In vivo, human case study	66,291	[46]
		*Pbunavirus E215*	vB_PaeM_E215	In vitro and vivo, human case study	66,789	[46]
		*Pbunavirus PA5*	PA5	Human case study	66,182	[47]
		*Pbunavirus PA10*	PA10	Human case study	91,212	[48]
		*Pbunavirus* *DSM 19872 (JG005)*	DSM 19872 (JG005)	In vitro and vivo	Not reported	[49]
		*Pbunavirus* *BrSP1*	BrSP1	In vitro	66,189	[50]
		*Pbunavirus* *DSM 22045 (JG024)*	DSM 22045 (JG024)	In vitro and vivo	66,275	[49]
		*Pbunavirus* *PB10*	PB10	In vitro and vivo	66,096	[51]
		*Pbunavirus* *PA19*	PA19	In vitro and vivo	87,921	[51]
		*Pbunavirus TH15*	TH15	In vitro and vivo	65,936	[26]
		*Pbunavirus LS1*	Pa204	Human case study	65,924	[41]
*Chimalliviridae*	*Phikzvirus*	*Phikzvirus OMKO1*	OMKO1	Human case study	281,755	[52]
		*Phikzvirus KTN4*	KTN4	In vitro and vivo	279,593	[53]
-	Uncl. *Caudoviricetes*	Uncl. *Caudoviricetes*	ΦPan70	In vitro	Not reported	[54]
*Megamimivirinae*	-	*Myovirus PP1450*	PP1450	In vitro	Not reported	[55]
		*Myovirus APIs*	APIs	Human case study	Not reported	[56]
		*Myovirus PP1777*	PP1777	In vitro and vivo	Not reported	[55]
		*Myovirus P1797*	PP1797	In vitro and vivo	Not reported	[55]
		*Myovirus PP1792*	PP1792	In vitro and vivo	Not reported	[55]
		*Pakpunavirus PAKP1*	PAK_P1	In vitro	93,198	[25]
		*Pakpunavirus PAKP4*	PAK_P4	In vitro and vivo	93,147	[25]
		*Pakpunavirus CAb1*	vB_PaeM_C2-10_Ab1	In vitro	92,777	[57]
-	*Nankokuvirus*	*Nankokuvirus KPP10*	KPP10	In vitro and vivo	88,322	[58]
*Cystoviridae*	*Cystovirus*	*Cystovirus phiYY*	phiYY	In vitro and vivo, human case study	13,514 (dsRNA)	[59]

**Table 2 viruses-16-01051-t002:** A summary of phage therapy against human infections caused by *P. aeruginosa*.

Infectious Syndrome	Patient Sex and Age (Years)	Phage(s)	Administration Route	Highest Phage Dosage (PFU/mL)	Length of the Phage Administration	Survived Initial Infection	Clinical Outcomes	Ref.
Chronic lung infection	Man, 40	phiYY	nebulized	1 × 10^9^	3 days	Yes	After 3 days of treatment, the patient was discharged with relieved symptoms of infection	[133]
Pneumonia and empyema	Female, 77	AB-PA01 (Pa 193, Pa 204, Pa 222, Pa 223)	i.v. and nebulized	1 × 10^9^	7 days	Yes	Clinical cure	[134]
Pneumonia in a lung transplant recipient	Female, 52	AB-PA01 (Pa193, Pa204, Pa222, and Pa223)	i.v.	4 × 10^9^	4 weeks	Yes	Recovered	[135]
Pneumonia in a lung transplant recipient	Male, 67	AB-PA01 (Pa193, Pa204, Pa222, and Pa223)	i.v. and nebulized	4 × 10^9^	29 days	Yes	Day 46 saw a second hospitalization for pneumonia; both pneumonia bouts were successfully treated	[135]
		AB-PA01 m1 (Pa176, Pa193, Pa223, Pa204, Pa222)	i.v. and nebulized	5 × 10^9^	46 days			
		Navy phage cocktail (PaSKWϕ17, Paϕ1 and PaSKWϕ22)	i.v. and nebulized	1 × 10^9^	93–150 days			
		Navy phage cocktail 2 (PaATFϕ1 and PaATFϕ3)	i.v.	5 × 10^7^	93–150 days			
CF exacerbation	Female, 26	AB-PA01 (Pa193, Pa204, Pa222, and Pa223)	i.v.	4 × 10^9^	8 weeks	Yes	Clinical progress: No CF exacerbations seven days into treatment, and no CF exacerbations for 100 days after phage therapy ended	[136]
Ventricular assist device infection	Male, 60	GD-1 (3 phages)	i.v.	1.9 × 10^7^	6 weeks	Failure	1 week after beginning phage therapy, developed bacteremia; after phage therapy ended, experienced recurring purulent discharge	[137]
Ventricular assist device infection	Male, 82	SDSU1 (2 phages: E217, PAK_P)	i.v. and direct application	7.58 × 10^5^	6 weeks	Failure	He received two rounds of phage therapy and one week after the first episode’s conclusion, he experienced recurrent bacteremia. After 3.5 months of starting phage therapy treatment, the patient had recurrent bacteremia four weeks into episode 2	[137]
		SDSU2 (2 phages: PAK_P5, PAK_P1)		4 × 10^10^	3 weeks			
		PPM3 (4 phages)		1 × 10^9^	4 weeks			
Recurrent bacteremia and probable aortic graft infection	Male, 64	PPM2 (3 phages)	i.v.	2.6 × 10^6^	6 weeks	Yes	For the previous 1.5 years, there has been recurrent bacteremia with extended antibiotic regimens and breakthrough infection. Blood cultures were negative when taking ciprofloxacin and phage therapy	[137]
Ventricular assisted device infection	Male, 53	APIs (PNM, 14/1, PT07)	i.v. and direct application	5 × 10^9^	5 days	Yes	Clinical cure; patient passed away 4 months later from non-infectious reasons.	[56]
Periprosthetic joint infection	Male, 88	PP1450, PP1777, and PP1792	Direct application	3 × 10^10^	Once	Yes	Resolution	[55]
Osteomyelitis	Male, 60	1777, 1792, 1797, 1450	Direct application	(1.2–9.7) × 10^8^	12 days	No	The wound’s appearance improved by day 14	[138]
Osteoarticular infection	Child, 7	Pa14NPΦPASA16 (PASA16)	i.v.	1.72 × 10^11^	2 weeks	Clinical success	Initiation of complementary phage therapy in just two weeks showed significant therapeutic benefits	[139]
Cardiothoracic surgery	Male, 13	PA5, PA10	locally	4 × 10^10^	_	Clinical success	After phage therapy, *P. aeruginosa* was not found and recovery was achieved	[48]
Vascular prosthesis	Male, 76	OMKO1	i.v.	10 × 10^7^	_	Clinical success	The infection was eradicated by a single administration of ceftazidime and the OMKO1 phage	[52]
Urinary tract	Male, 61	Bacteriophage cocktail BFC1. PNM bacteriophages, Bacteriophage 14/1, ISP bacteriophages (*S. aureus*)	i.v.	50 mL of BFC1 bacteriophage cocktail	_	Clinical success	Avoiding hemofiltration and the absence of any unforeseen side effects	[140]
Leg ulcers	39 patients	WPP-201, a cocktail of 8 lytic bacteriophages	Topically	1 × 10^9^	12 weeks	No	There was no appreciable difference in the occurrence of negative events or healing between the phage-treated groups and the control group	[141]

## Data Availability

Not applicable.

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
