# Peer review of "Pseudomonas aeruginosa Bacteriophages and Their Clinical Applications"

_viruses, 2024, doi:10.3390/v16071051_

Round 1
Reviewer 1 Report
Comments and Suggestions for Authors
The manuscript by Alipour-Khezri et al. should concern Pseudomonas aeruginosa bacteriophages and their clinical application, as indicated by the manuscript title. The authors put a lot of effort into collecting the data provided and in general the manuscript contains valuable information. However, the manuscript text is chaotic and will fit the title only when reorganized, shortened substantially to remove irrelevant data, and supplemented with additional data. In chapter 2 the authors should not limit the description of phages to show their morphotypes and a few examples of taxonomic classification. Phages of similar morphotypes can have very different properties and different lifestyles. Thus, I would suggest supplementing Chapter 2 with an additional table containing a more detailed description of P. aeruginosa phages used in therapies. The relevant data can be found on the home page of the International Committee on Taxonomy of Viruses (ICTV; https://ictv.global/). Current Table 1 could become Table 2 but without the column concerning phage morphotype. Additionally, in certain parts of the manuscript text, it is unclear which results cited by the authors apply to phages in general, and which of them apply to P. aeruginosa phages. The authors should separate the information in the text accordingly, and remove excess data concerning phages other than those infecting P. aeruginosa.
The manuscript could be substantially shortened to retain the logical flow of topics in the text, without providing too many details that distract the readers' attention. For example, the descriptions of certain therapy trials are too detailed and too unclear as for the review article (see e.g. L. 280-315). A more general summary of methods and results would be sufficient. If possible, the authors could compare the therapeutic efficacy of various P. aeruginosa phages.
Chapter 4 requires rewriting, for precision and consistency (see further below),
L. 56: Please italicize "in vivo" and "in vitro", also elsewhere in the manuscript
L. 76: Replace "new order" with "class"
L. 116: please italicize "Enterobacteriaceae"
L. 119 and elsewhere in the text: Replace "gram-negative" with "Gram-negative". The method is named after the name of its discoverer.
L. 134-133: Please add, that the experiment was performed in a porcine infection model (to avoid confusion).
L. 185-188: Please provide a reference(s).
L. 197-198: Delete the sentence. It is irrelevant to the manuscript's focus.
L. 238-239: Replace "Bacterial" with "P. aeruginosa" to conform to the content of the cited publication.
L. 253: What do you mean by "immediately lethal"? Could you replace it with a more commonly used or more precise term?
L. 329-331: Rewrite the sentence, for clarity. What do you mean by "local induction of phages"?
L. 339-341: This sentence does not apply to P. aeruginosa infection and should be removed.
L. 351-352: Rewrite the sentence, for clarity. what do you mean by "altered on a regular basis"? Be more specific.
L. 355-356: Rewrite the sentence for better binding to its part in parentheses.
L. 357: This is unclear. Why phage therapeutics are to be genetically altered to be used in vitro? What is the difference between (4) and (1)?
L. 360: "Phages, like other proteins,"? Phages are not proteins. They contain proteins.
L. 375: Please, be more precise.
L. 378-383: Be more precise. In another case, it is difficult to understand what exactly was done.
L. 387: Genetic tools were rather developed, not simply discovered.
L. 682: "unintentionally"? This is unjustified anthropomorphism.
Additionally, the manuscript contains spelling errors, see, e.g., " intravesical hage administration" (L. 533).
Comments on the Quality of English Language
This manuscript requires rewriting, substantial abbreviation, and better ordering of particular chapters in the text. In its current form, it is chaotic and difficult to read.
Author Response
Reviewer #1
The manuscript by Alipour-Khezri et al. should concern Pseudomonas aeruginosa bacteriophages and their clinical application, as indicated by the manuscript title. The authors put a lot of effort into collecting the data provided and in general the manuscript contains valuable information. However, the manuscript text is chaotic and will fit the title only when reorganized, shortened substantially to remove irrelevant data, and supplemented with additional data.
- In chapter 2 the authors should not limit the description of phages to show their morphotypes and a few examples of taxonomic classification. Phages of similar morphotypes can have very different properties and different lifestyles. Thus, I would suggest supplementing Chapter 2 with an additional table containing a more detailed description of aeruginosa phages used in therapies. The relevant data can be found on the home page of the International Committee on Taxonomy of Viruses (ICTV; https://ictv.global/).
Authors’ reply: Thank you for your constructive comments. We have added a new table to chapter 2 as suggested The new Table 1 contains details of P. aeruginosa phages used for therapeutic purposes.
- Current Table 1 could become Table 2 but without the column concerning phage morphotype.
Authors’ reply: Current table is new Table 3, as the order of original tables had to be changed. We have omitted the phage morphotype columns from both tables.
- Additionally, in certain parts of the manuscript text, it is unclear which results cited by the authors apply to phages in general, and which of them apply to aeruginosa phages. The authors should separate the information in the text accordingly, and remove excess data concerning phages other than those infecting P. aeruginosa.
Authors’ reply: The whole text has been checked and revised according to the request.
- The manuscript could be substantially shortened to retain the logical flow of topics in the text, without providing too many details that distract the readers' attention. For example, the descriptions of certain therapy trials are too detailed and too unclear as for the review article (see e.g. L. 280-315). A more general summary of methods and results would be sufficient. If possible, the authors could compare the therapeutic efficacy of various aeruginosa phages.
Authors’ reply: We have revised and shortened the manuscript to retain the logical flow of topics as recommended. The therapeutic efficacy of various phages is demonstrated for the patient cases presented in table 3. L. 280-315, corrected.
- Chapter 4 requires rewriting, for precision and consistency (see further below),
Authors’ reply: Chapter 4 has been revised.
3- L. 56: Please italicize "in vivo" and "in vitro", also elsewhere in the manuscript.
Response: Revised as requested.
4- L. 76: Replace "new order" with "class".
Response: Replaced.
5- L. 116: please italicize "Enterobacteriaceae".
Response: Revised as requested.
6- L. 119 and elsewhere in the text: Replace "gram-negative" with "Gram-negative". The method is named after the name of its discoverer.
Response: Revised as requested.
7- L. 134-133: Please add, that the experiment was performed in a porcine infection model (to avoid confusion).
Response: Revised as requested, the phrase "conducted on a porcine infection model" was added to the text.
8- L. 185-188: Please provide a reference(s).
Response: Added as requested.
9- L. 197-198: Delete the sentence. It is irrelevant to the manuscript's focus.
Response: Revised as requested.
10- L. 238-239: Replace "Bacterial" with "P. aeruginosa" to conform to the content of the cited publication.
Response: Revised as requested.
11- L. 253: What do you mean by "immediately lethal"? Could you replace it with a more commonly used or more precise term?
Response: The word "immediately" was replaced with "highly".
12- L. 329-331: Rewrite the sentence, for clarity. What do you mean by "local induction of phages"?
Response: The word " induction " was replaced with "injection".
13- L. 339-341: This sentence does not apply to P. aeruginosa infection and should be removed.
Response: Revised as requested.
14- L. 351-352: Rewrite the sentence, for clarity. what do you mean by "altered on a regular basis"? Be more specific.
Response: Revised as requested.
15- L. 355-356: Rewrite the sentence for better binding to its part in parentheses.
Response: Revised as requested.
16- L. 357: This is unclear. Why phage therapeutics are to be genetically altered to be used in vitro? What is the difference between (4) and (1)?
Response: Revised as requested.
17- L. 360: "Phages, like other proteins,"? Phages are not proteins. They contain proteins.
Response: Revised as requested.
18- L. 375: Please, be more precise.
Response: Revised as requested.
19- L. 378-383: Be more precise. In another case, it is difficult to understand what exactly was done.
Response: Revised as requested.
20- L. 387: Genetic tools were rather developed, not simply discovered.
Response: The word "discovery " was replaced with "developing".
21- L. 682: "unintentionally"? This is unjustified anthropomorphism.
Response: The word " unintentionally" was replaced with " inadvertently ".
22- Additionally, the manuscript contains spelling errors, see, e.g., " intravesical hage administration" (L. 533).
Response: Revised as requested.
Comments on the Quality of English Language
This manuscript requires rewriting, substantial abbreviation, and better ordering of particular chapters in the text. In its current form, it is chaotic and difficult to read.
Response: The whole text has been revised to make it read better.
Reviewer 2 Report
Comments and Suggestions for Authors
This manuscript of Elaheh Alipour-Khezri and and colleagues provides an overview of recent advances in the use of bacteriophages infecting Pseudomonas aeruginosa to treat diseases and diseases, as well as the results of these treatments. With the growing problems associated with multidrug resistance, phage therapy is becoming a possible alternative or complement to antibiotic therapy for infectious diseases, especially nosocomial infections. The authors reviewed published sources and analyzed nearly two hundred articles, collecting and organizing data on treatment options, phages used, and forms of phage use in treatment. The structure of the manuscript is logical, the data is presented convincingly and well discussed. However, the manuscript could benefit from addressing some minor issues.
Section 2. It would be interesting to describe the general requirements for phages used for phage therapy, including lytic lifestyle and gene composition.
Table 1. Please remove italics from the “Myovirus”, unnecessary dots after the names of morphological types (phage SaPL) and in the description of the outcome.
In addition, the manuscript would have benefited from a brief discussion of problems such as the emergence and development of resistance to phages, sometimes too narrow a lytic spectrum, and possible solutions to these problems.
Author Response
Reviewer #2
1- Section 2. It would be interesting to describe the general requirements for phages used for phage therapy, including lytic lifestyle and gene composition.
Response: A sentence on that was added to chapter 2
2- Table 1. Please remove italics from the “Myovirus”, unnecessary dots after the names of morphological types (phage SaPL) and in the description of the outcome.
Response: Revised as requested. The table is new Table 3
3- In addition, the manuscript would have benefited from a brief discussion of problems such as the emergence and development of resistance to phages, sometimes too narrow a lytic spectrum, and possible solutions to these problems.
Response: The issue has been briefly discussed in the revised manuscript
Reviewer 3 Report
Comments and Suggestions for Authors
Dear authors,
the topic of your article is interesting and actual. The tables help to orientate in the field and are valuable for the readers. The readers might appreciate adding information about the ongoing clinical trials. The overall quality of the review is good. The text contains a minimum of typos and I found no major issue. I would recommend the changes below.
Line 553: Please, change „hage“ to „phage".
Table 2: The second column/first row - "Patient de-mographic s" - please, correct the hyphenation of the word.
The chapter order in the case of chapters 7, 8 and 9 seems random. I would suggest organizing chapters 7 and 9 into one larger chapter called, e.g., "P. aeruginosa and community-onset infections".
Line 634: I would suggest to erase the word "virus".
Lines 653-655: The sentence is unclear.
Line 656: Please, change the font size in "P. aeruginosa", it is bigger than the surrounding text.
Line 677: Please, change "Myovirus of genus Pakpunavirus" to "myovirus of genus Pakpunavirus".
Line 686: There is a double gap.
Author Response
Reviewer #3
1- Line 553: Please, change „hage“ to „phage".
Response: Revised as requested
2- Table 2: second column/first row - "Patient de-mographic s" - please, correct the hyphenation of the word.
Response: The table has been revised
3- The chapter order in the case of chapters 7, 8 and 9 seems random. I would suggest organizing chapters 7 and 9 into one larger chapter called, e.g., "P. aeruginosa and community-onset infections".
Response: Revised as requested
4- Line 634: I would suggest to erase the word "virus".
Response: Revised as requested
5- Lines 653-655: The sentence is unclear.
Response: Revised.
6- Line 656: Please, change the font size in "P. aeruginosa", it is bigger than the surrounding text.
Response: Revised as requested
7- Line 677: Please, change "Myovirus of genus Pakpunavirus" to "myovirus of genus Pakpunavirus".
Response: Revised as requested
8- Line 686: There is a double gap.
Response: Revised as requested
Round 2
Reviewer 1 Report
Comments and Suggestions for Authors
The revised version of the manuscript by Alipour-Khezri et al. contains certain improvements compared to the previous version, but the manuscript still requires further editing, to make it less messy and easier to read. The authors changed the order of certain parts but did not sufficiently shorten the manuscript by better grouping all its threads to avoid their repetitions. Still, too little attention was paid to the description of Pseudomonas therapeutic phages. The authors show in Table 1 that Pseudomonas phages used in therapies belong to six families or subfamilies and 11 genera within these families or subfamilies. However, they did not mention any differences or similarities or other characteristic features of phages of particular taxons, like, e.g., at least the ranges of genome sizes, morphotype, phage lifestyle (exclusively lytic or lysogenic) kinds of preferred receptors (if known), etc. The title suggests a wider description than that provided in the current version of the manuscript. Additionally, the logic of the manuscript will benefit from additional reorganizations. Why not put into a separate chapter the description of all kinds of infections caused by P. aeruginosa as well as the therapeutic problems associated with them, then describe benefits of using phages in therapies in general, then provide information about forms of phage delivery, and at the end provide examples of the application of phages to cure P. aeruginosa caused infections. In the current version of the manuscript, many threads are unnecessarily repeated in different parts, and treads concerning certain aspects associated with phages and phage therapy in general are mixed with aspects concerning specifically phage therapy of Pseudomonas aeruginosa infections. It will be easier to complete the list of specific comments if the authors organize the manuscript in a better way. I realize that this is a difficult task because of the complexity of the manuscript subject, but it is worth doing to improve the clarity of the information provided in the manuscript. Only some of my specific comments are below.
Table 1: The names of phage families, subfamilies, genera, and species should be italicized. Information about each phage used in therapy should be supplemented with the relevant references in a separate column of this table.
L. 108: "gram-negative" See the comments to the previous version of the manuscript.
L. 148-149: The logical connection between the sentences is unclear. They should be rewritten to provide a logical connection between them.
L. 161-176: This text should be in Chapter 4. In another case, it disrupts the logic of Chapter 3.3.
Table 2: Please indicate that it is about phage therapy of P. aeruginosa infections in humans. Include Table 3 before Table 2 in the manuscript to fit the order of tables to the order of chapters in the manuscript text.
Chapter 5 should be rather entitled "Phage Therapy in Humans - Progress and Limitations". Examples of phage therapy in humans are spread in other parts of the manuscript too.
Author Response
Replies to the comments of reviewer #1
The revised version of the manuscript by Alipour-Khezri et al. contains certain improvements compared to the previous version, but the manuscript still requires further editing, to make it less messy and easier to read.
- The authors changed the order of certain parts but did not sufficiently shorten the manuscript by better grouping all its threads to avoid their repetitions. HAVE BEEN REVISED
- Still, too little attention was paid to the description of Pseudomonas therapeutic phages. The authors show in Table 1 that Pseudomonas phages used in therapies belong to six families or subfamilies and 11 genera within these families or subfamilies. However, they did not mention any differences or similarities or other characteristic features of phages of particular taxons, like, e.g., at least the ranges of genome sizes, morphotype, phage lifestyle (exclusively lytic or lysogenic) kinds of preferred receptors (if known), etc. The title suggests a wider description than that provided in the current version of the manuscript. ADDITIONAL INFORMATION HAS BEEN ADDED TO THE REVISED MANUSCRIPT
- Additionally, the logic of the manuscript will benefit from additional reorganizations. HAS BEEN REORGANIZED
- Why not put into a separate chapter the description of all kinds of infections caused by P. aeruginosa as well as the therapeutic problems associated with them, DONE AS SUGGESTED
- then describe benefits of using phages in therapies in general, DONE
- then provide information about forms of phage delivery, DONE
- and at the end provide examples of the application of phages to cure P. aeruginosa caused infections. DONE
- In the current version of the manuscript, many threads are unnecessarily repeated in different parts, and treads concerning certain aspects associated with phages and phage therapy in general are mixed with aspects concerning specifically phage therapy of Pseudomonas aeruginosa infections. REORGANIZED TO GIVE A BETTER FLOW TO THE TEXT
- It will be easier to complete the list of specific comments if the authors organize the manuscript in a better way. I realize that this is a difficult task because of the complexity of the manuscript subject, but it is worth doing to improve the clarity of the information provided in the manuscript. WE HAVE REVISED THE MANUSCRIPT SUBSTANTIALLY FOLLOWING THE POINTS ABOVE. ALL THE CHANGES IN THE TEXT HAVE BEEN HIGHLIGHTED IN YELLOW.
Only some of my specific comments are below.
Table 1: The names of phage families, subfamilies, genera, and species should be italicized. Information about each phage used in therapy should be supplemented with the relevant references in a separate column of this table. DONE
- 108: "gram-negative" See the comments to the previous version of the manuscript. DONE
- 148-149: The logical connection between the sentences is unclear. They should be rewritten to provide a logical connection between them. REWRITTEN
- 161-176: This text should be in Chapter 4. In another case, it disrupts the logic of Chapter 3.3. REVISED
Table 2: Please indicate that it is about phage therapy of P. aeruginosa infections in humans. Include Table 3 before Table 2 in the manuscript to fit the order of tables to the order of chapters in the manuscript text. THE ORDER OF TABLES HAS BEEN CHECKED
Chapter 5 should be rather entitled "Phage Therapy in Humans - Progress and Limitations". Examples of phage therapy in humans are spread in other parts of the manuscript too. REVISED